# Co-Delivery of Repurposing Itraconazole and VEGF siRNA by Composite Nanoparticulate System for Collaborative Anti-Angiogenesis and Anti-Tumor Efficacy against Breast Cancer

**DOI:** 10.3390/pharmaceutics14071369

**Published:** 2022-06-28

**Authors:** Mingji Jin, Bowen Zeng, Yanhong Liu, Lili Jin, Yan Hou, Chao Liu, Wei Liu, Hao Wu, Liqing Chen, Zhonggao Gao, Wei Huang

**Affiliations:** 1State Key Laboratory of Bioactive Substance and Function of Natural Medicines, Institute of Materia Medica, Chinese Academy of Medical Sciences and Peking Union Medical College, Beijing 100050, China; jinmingji@imm.ac.cn (M.J.); 2020010524@ybu.edu.cn (B.Z.); liuyanhong@imm.ac.cn (Y.L.); houyan987654@aliyun.com (Y.H.); chaoliu@imm.ac.cn (C.L.); liuw@imm.ac.cn (W.L.); wuhao1230@aliyun.com (H.W.); chenliqing@imm.ac.cn (L.C.); 2Beijing Key Laboratory of Drug Delivery Technology and Novel Formulations, Institute of Materia Medica, Chinese Academy of Medical Sciences and Peking Union Medical College, Beijing 100050, China; 3Department of Respiratory Medicine, Yanbian University Hospital, Yanji 133000, China; 4Department of Pharmacy, Yanbian University, Yanji 133000, China; jinlili@ybu.edu.cn

**Keywords:** itraconazole, VEGF siRNA, siRNA delivery, breast cancer, co-inhibition

## Abstract

Combinations of two different therapeutic modalities of VEGF inhibitors against angiogenesis can cooperatively impede breast cancer tumor growth and enhance therapeutic efficacy. Itraconazole (ITZ) is a conventional antifungal drug with high safety; however, it has been repurposed to be a multi target anti-angiogenesis agent for cancer therapy in recent years. In the present study, composite nanoparticles co-loaded with ITZ and VEGF siRNA were prepared in order to investigate their anti-angiogenesis efficacy and synergistic anticancer effect against breast cancer. The nanoparticles had a suitable particle size (117.9 ± 10.3 nm) and weak positive surface charge (6.69 ± 2.46 mV), as well as good stability and drug release profile in vitro. Moreover, the nanoparticles successfully escaped from endosomes and realized cell apoptosis and cell proliferation inhibition in vitro. In vitro and in vivo experiments showed that the nanoparticles could induce the silencing of VEGF-related expressions as well as anti-angiogenesis efficacy, and the co-loaded ITZ-VEGF siRNA NPs could inhibit tumor growth effectively with low toxicity and side effects. Taken together, the as-prepared delivery vehicles are a simple and safe nano-platform that improves the antitumor efficacy of VEGF siRNA and ITZ, which allows the repositioning of the generic drug ITZ as a great candidate for antitumor therapy.

## 1. Introduction

Combination therapy with anticancer drugs and siRNA has been suggested to be an effective and synergistic strategy for the cancer treatment with advantages of enhancing therapeutic effects [1,2]. However, the combined delivery of anticancer drugs and siRNA remains challenging due to their different physical and chemical properties [3]. In our previous study, we designed and prepared a novel smart polymeric nanoparticle delivery system using polyethyleneimine-block-polylactic acid (PEI-PLA)/poly (ethylene glycol)-block-poly (L-aspartic acid sodium salt) (PEG-PAsp) to co-deliver small molecule drugs and siRNA [4,5]. 

The prepared complex nanoparticles co-loaded with chemotherapeutic drugs and siRNA exhibited a good synergistic effect and inhibitory effect on both non-small cell lung cancer and breast cancer. In addition, this kind of carrier system has the advantages of easy preparation, high stability and low toxicity, and it is necessary and important to expand its application range.

Drug repurposing represents the identification of the novel pharmacological effects of conventional drugs, which is also cost-effective and time saving [6]. As the pharmacokinetics, pharmacodynamics and safety of traditional drugs have been established, expanding the application of this drug in other diseases is also a rapid and risk-free way to develop new drug. Itraconazole (ITZ) is a broad-spectrum antifungal drug with triazole ring, of which the mechanism is to inhibit the synthesis of ergo sterol mediated by cytochrome P450 oxidase and change the permeability of fungal cell membrane [7]. 

In recent years, researchers have found that ITZ has antitumor effect, which has changed people’s understanding of its traditional identity as an antifungal agent [8]. Preclinical and clinical evidence confirmed the activity of ITZ against different cancers, rationalizing its potential repurposing as a chemotherapeutic agent [6]. Wang [9] et al. administered ITZ to MCF-7 and SKBR-3 nude mice bearing human breast cancer and observed its antitumor effects in vitro and in vivo. 

The results showed that ITZ could induce tumor volume reduction and promoted apoptosis and autophagy in tumor bearing mice. Some studies [10] have found that ITZ can block the growth of tumor vascular endothelial cells in G1 phase, and then inhibit tumor angiogenesis. Interestingly, ITZ has a high affinity for fungal cytochrome P450 and a low affinity for human cytochrome P450 [7,11], thus, being considered to be a potential antitumor drug with low toxicity.

Vascular endothelial growth factor (VEGF) is a member of the angiogenic factor family, which is also a key point in promoting angiogenesis [12]. Angiogenesis is a type of excessive proliferation of irregular blood vessels in the tumor microenvironment, which occurs in a wide spectrum of cancers [13,14]. Growing tumors require new blood vessels to supply nutrients and oxygen [15,16], and the non-angiogenic tumors require co-opting pre-existing vessels to induce or access vasculature [17,18], and thus angiogenesis inhibitors have been proposed as anticancer drugs. 

As a factor with high expression in most solid tumors, VEGF silencing can cause the apoptosis of vascular endothelial cells, and blocking VEGF can directly inhibit the growth of tumors [19,20]. Since VEGF is identified as the ideal RNAi candidate for breast cancer treatment, RNAi-mediated silencing of VEGF has demonstrated a great capability for VEGF expression inhibition [21]. As the progression of breast cancer is more intensely dependent on angiogenesis than others, a dual-targeting approach against angiogenesis and breast cancer cell proliferation would synergistically enhance the therapeutic efficacy [22].

The combination of ITZ and other drugs may be a potential therapy for breast cancer. By studying the anticancer mechanism of ITZ, Nacev and others [23] confirmed that ITZ could significantly inhibit the binding of VEGF and VEGF receptor 2 (VEGFR2). The combination of ITZ and VEGF monoclonal antibodies also exerted a better synergistic effect. Hara and others [24] found that ITZ and bevacizumab, a monoclonal antibody with anti VEGF function, played a synergistic role in anti-angiogenesis. 

These theories further enrich the mechanism of ITZ in antitumor vascular proliferation and provide the basis for tumor treatment. Therefore, ITZ co-loaded with VEGF siRNA would be a good strategy for anti-angiogenesis chemotherapy. However, ITZ has a highly hydrophobic weak base group with low solubility in water (about 1 ng/mL) [25,26], while VEGF siRNA is a hydrophilic high-molecular-weight drug. The most challenging problem is the efficient delivery of both ITZ and VEGF siRNA into specific target tissues without toxic side effects.

Our team’s previous research already confirmed that the PEI-PLA and PEG-PAsp delivery system is effective for co-loading anticancer drugs and siRNAs. In further research on synergistic effect of anti-angiogenesis chemotherapy, we continue to choose PEI-PLA and PEG-PAsp as a drug delivery system to co-load the extremely hydrophobic ITZ and hydrophilic VEGF siRNA (Figure 1a). The prepared complex nanoparticles are expected to be stable in the blood stream (pH 7.4) and tumor extracellular environment (pH 6.5). Once entering tumor cells via EPR effect, the PEG-PAsp block will become neutral in acid endosomal environment (pH 5.0–6.0) and therefore detach from the complex nanoparticles. Finally, lysosomal escape occurred due to the “proton sponge effect” of PEI, and the drugs were successfully released to tumor cells. Under the combined effect of the two types of drugs, the co-loaded nanoparticles can synergistically inhibit tumor angiogenesis and thus play an antitumor role against breast cancer (Figure 1b).

## 2. Materials and Methods

### 2.1. Materials

Poly (lactic acid) with carboxyl groups on one end (PLA-COOH, Mw = 5–15 KDa) was purchased from Jinan Daigang Technology Co., Ltd. (Jinan, China). Branched polyethyleneimine (Mw = 1.8 KDa, bPEI_1.8k_) was purchased from Alfa Aesar (Ward Hill, MA, USA). Methoxyl-poly (ethylene glycol)-block-poly (L-aspartic acid sodium salt) (PEG-PAsp, Mw = 6.4 KDa) was purchased from Alamanda Polymers (Huntsville, AL, USA). 1-(3-Dimethylaminopropyl)-3-ethylcarbodiimide hydrochloride (EDC) and N-hydroxysuccinimide (NHS) were obtained from Sigma-Aldrich Inc. (Shanghai, China). Itraconazole (ITZ) was purchased from Sigma-Aldrich Inc. (Shanghai, China). Coumarin-6 (C-6) was purchased from J&K Scientific. (Beijing, China). RNase A was purchased from Solarbio Ltd. (Beijing, China). TRIzol was purchased from Invitrogen Company (Carlsbad, CA, USA). 4′, 6-Diamidino-2-phenylindole (DAPI) and Hoechst 33258 were bought from the Beyotime Institute of Biotechnology (Shanghai, China). 

FITC-Annexin V/PI apoptosis detection kit was purchased from KeyGEN biosciences company (Nanjing, China). A BCA protein concentration determination kit (BCA) was purchased from Tiangen Biological Technology Co. (Beijing, China). Trypsin, HEPES buffer, PBS, DMEM and RPMI-1640 media were obtained from Thermo Fisher Scientific Co., Ltd. (Beijing, China). An Angiogenesis Assay Kit was purchased from Abcam (ab204726; Shanghai, China). A cell counting kit-8 (CCK-8) was obtained from Dojindo Laboratories (Kumamoto, Japan). SiRNA, targeting VEGF: 5′-CGAUGAAGCCCUGGAGUGCdTdT-3′ (sense), negative control siRNA (siNC): 5′-UUCUCCGAACGUGUCACGUTT-3′ (sense) and Cy3 siRNA were purchased from GenePharma Co., Ltd. (Shanghai, China). All other reagents were of analytical grade.

The 4T1 cells were acquired from the Department of Pathology in the Institute of Medicinal Biotechnology at Peking Union Medical College, a stable luciferase transfected cell line (4T1^Luc^) was constructed by our laboratory. They were grown in RPMI 1640 media with 10% fetal bovine serum (FBS) at 37 °C in a 5% CO_2_ atmosphere.

The Human Umbilical Vein Endothelial Cells (HUVEC) were acquired from the Department of Pathology in the Institute of Medicinal Biotechnology at Peking Union Medical College. They were grown in DMEM media with 10% fetal bovine serum (FBS), penicillin (IU/mL) and streptomycin (100 µg/mL) at 37 °C in a 5% CO_2_ atmosphere. 

Female BALB/c nude mice (4–6 weeks old, 18–22 g) were acquired from Vital River Laboratory Animal Technology Co. (Beijing, China). All animal studies were approved by the Laboratory Animal Ethics Committee in the Institute of Materia Medica at the Chinese Academy of Medical Sciences (CAMS) and Peking Union Medical College (PUMC). All the experimental procedures were performed in conformity with the institutional guidelines and protocols for the care and use of laboratory animals.

### 2.2. Preparation of ITZ-siRNA^VEGF^ NPs

The conjugation of PLA-COOH and bPEI_1.8k_, was synthesized according to the method previously reported by our group [4]. Briefly, 1300 mg of PLA-COOH (1 mmol) was dissolved in DMSO (5 mL), and then EDC (5 mmol) and NHS (5 mmol) were added and stirred at room temperature for 2 h. bPEI_1.8k_ (0.33 mmol) was added into the dimethyl sulfoxide (DMSO) solution and stirred for another 24 h in the room temperature. The reacted mixture was dialyzed (MWCO 3500) against 50% alcohol and distilled water, respectively at room temperature for 2 days to remove the extra products. The PEI-PLA copolymer was obtained after freeze drying. The structure of PEI-PLA was characterized by ^1^H NMR spectroscopy (Varian Mercury-600 MHz spectrometer, Varian Medical Systems, Inc., Palo Aito, CA, USA) using D_2_O as solvent and were further confirmed by FTIR (Nicolet 5700, Thermo Fisher Scientific, Beijing, China).

The ITZ-loaded NPs (ITZ NPs) were prepared by the dialysis method [27]. ITZ (5 mg) and PEI-PLA (50 mg) dissolved in 2 mL DMSO were added dropwise to 20 mL of water under stirring. The mixture was stirred for another 30 min at room temperature and dialyzed against distilled water using 7 KDa dialysis bag for 24 h. The unentrapped ITZ was removed by filtration through 0.45 µm filter (GE Healthcare, Boston, MA, USA), and the filtered solution was freeze-dried [28]. For the preparation of VEGF siR-nanoparticles (siRNA^VEGF^ NPs), PEI-PLA was diluted with distilled water to a certain concentration at N/P ratios (molar ratio of amino groups of PEI to phosphorus groups of siRNA) of 30 and then mixed with equal volumes of siRNA solutions (concentration of 2 pmol/µL). 

After vortexing for 5 s, the mixture was kept at room temperature for 20 min to form the siRNA^VEGF^ NPs. The ITZ and VEGF siRNA co-loaded nanoparticles (ITZ-siRNA^VEGF^ NPs) were prepared in the same way (N/P = 30). Finally, PEG-PAsp was diluted with distilled water to a certain concentration at C/N ratio (molar ratio of carboxyl groups of PEG-PAsp to amino groups of PEI-PLA) of 1/5, added to the above prepared solutions. After incubation at room temperature for 20 min, the final nanoparticles were prepared. The particle sizes and zeta potentials of the prepared nanoparticles were measured at 25 °C using Malvern Zetasizer Nano ZS90 (Malvern instruments Ltd., Worcestershire, UK). The morphology of siRNA^VEGF^ NPs was observed using transmission electron microscopy (TEM, Hitachi H-7650, Hitachi Ltd., Tokyo, Japan) at a voltage of 80 kV.

### 2.3. Determination of Drug-Loading Efficiency (DL) and Encapsulation Efficiency (EE) in ITZ-siRNA^VEGF^ NPs by HPLC

The following equations were used to calculate the DL and EE. The ITZ concentration was detected by an Agilent 1200 LC (Agilent Tech, Santa Clara, CA, USA) HPLC system using an Inertsustain C_18_ column (5 µm, 4.6 mm × 250 mm). The mobile phase consisted of acetonitrile and water (75:25, *v*/*v*) delivered at a flow rate of 1.0 mL/min. The injection volume was 20 µL, and the wavelength was set at 262 nm [29].
$$\mathrm{DL}\;=\frac{Amount\;of\;\mathrm{ITZ}\;in\;NPs}{Amount\;of\;\mathrm{ITZ}-loaded\;NPs}\;\times \;100\%\;\;\mathrm{EE}\;=\frac{Amount\;of\;\mathrm{ITZ}\;in\;NPs}{Amount\;of\;\mathrm{ITZ}\;for\;loading}\;\times \;100\%\;$$

### 2.4. Gel Retardation Assay of ITZ-siRNA^VEGF^ NPs

The siRNA association was evaluated by the gel retardation assay on 4% agarose gel, and the electrophoresis was performed at 120 V for 20 min. Subsequently, the gel was stained with 0.5 mg/mL EtBr for 30 min and photographed under an UV image system (SIM135A, SIMON).

### 2.5. Serum Stability of ITZ-siRNA^VEGF^ NPs

To evaluate the serum stability, ITZ-siRNA^VEGF^ NPs were incubated at 37 °C in PBS supplemented with or without 10% fetal bovine serum (FBS), respectively. The average particle sizes of the nanoparticles were monitored by dynamic light scattering (DLS) over a period of 24 h. Each sample was performed in triplicate.

### 2.6. In Vitro pH-Sensitive Ability of ITZ-siRNA^VEGF^ NPs

For the pH-sensitive analysis, the ITZ-siRNA^VEGF^ NPs were kept in 10 nM HEPES buffer of different pH values, respectively. Then, the particle sizes and zeta potentials of the co-loaded NPs were measured at 25 °C using Malvern Zetasizer Nano ZS90 (Malvern instruments Ltd., Worcestershire, UK).

In vitro pH-sensitive release of ITZ in different formulations was evaluated using dialysis diffusion technique. The release study was performed in PBS (pH 5.5 and pH 7.4) containing sodium lauryl sulphate (0.5%, *w*/*v* solution) up to 72 h as per the reported methods [30]. Each sample of 0.5 mL NPs containing 0.3 mg ITZ was added into a dialysis bag (7 KDa) and tightly sealed. Then, the bags were immersed in 40 mL PBS solution and incubated in an orbital shaker at 37 °C. At predetermined time points, 0.2 mL of each sample was withdrawn from the medium, and the same volume of fresh medium was added. Each sample was centrifuged at 10,000 r/min, and the supernatant was assayed by HPLC.

### 2.7. Confocal Laser Scanning Microscopy Analysis

To assess the cellular uptake and endosomal escape of the nanoparticles, confocal laser scanning microscopy (CLSM) was used. 4T1 cells were seeded onto coverslips in a 12-well plate at a density of 5 × 10^4^ cells per well and incubated at 37 °C to allow cell attachment. After 24 h, the medium was replaced with serum-free cell culture containing various formulations of NPs (N/P = 30, C/N = 1/5; C-6: 0.1 µg/mL) at 50 nM siRNA per well for 4 h. After washing with cold PBS, cells were fixed with 4% paraformaldehyde for 15 min. DAPI was subsequently added to stain the nuclei. Finally, the sample was observed and imaged on a confocal microscope (Carl Zeiss LSM 710, Carl Zeiss Microscopy GmbH, Wetzlar, Germany).

For endosomal escape investigation, 24 h cultured 4T1 cells on petri dish were cultured with siRNA^VEGF^-C6 NPs and siRNA^Cy5^-ITZ NPs for 1, 2, 4 and 6 h, respectively. The concentration of C6 and FAM siRNA was 1 µM and 100 nM, respectively. Then, the endosome was stained with Lyso-Tracker Red and Lyso-Tracker Green at 37 °C for 30 min. Subsequently, the cells were rinsed with cold PBS, fixed with 4% (*w*/*v*) formaldehyde and stained with DAPI. After washing away the residual dye, the endosome escape of the nanoparticles was observed on CLSM.

### 2.8. Quantitative Cell Uptake Study

To qualitatively evaluate the cellular uptake of co-loaded nanoparticles, they were labeled with Cy3 siRNA and C6. 4T1 cells were seeded into a 12-well plate at a density of 5 × 10^4^ cells per well and incubated at 37 °C to allow cell attachment. After 24 h, the medium was replaced with serum-free cell culture medium containing various formulations of NPs (N/P = 30, C/N = 1/5; C6: 0.5 µM) at 50 nM siRNA per well for 4 h. For the flow cytometry analysis, the cells were washed three times with PBS buffer and trypsinized, the harvested cells were resuspended in the fresh medium and washed with cold PBS. Finally, the cells were resuspended in 0.5 mL PBS buffer and analyzed with FACSCalibur flow cytometer (Becton Dickinson, Franklin Lake, NJ, USA).

### 2.9. Wound Healing Assay

4T1 cells were seeded into a 12-well plate at a density of 5 × 10^4^ cells per well for 24 h. Then, the confluent cell monolayer per well was wounded with a 200 μL pipette tip, washed with serum-free medium and exposed to different formulations (PBS, siRNA^VEGF^, ITZ, blank NPs, siRNA^VEGF^ NPs, ITZ NPs and ITZ-siRNA^VEGF^ NPs (N/P = 30; C/N = 1/5; ITZ content: 8.28%; 100 nM of siRNA/each well)), cells untreated were used as controls. Medium per well was replaced with fresh complete medium after 4 h incubation. The healing status of scratch wound were observed and imaged during the next culture time.

### 2.10. In Vitro Gene Silencing Efficiency Assay

To test the siRNA silencing efficacy on VEGF expression, 4T1 cells were incubated into a 6-well plate at a density of 1 × 10^5^ cells per well for 24 h. Then, the cells were incubated in 2 mL of serum-free RPMI 1640 medium containing different nanoparticles for 4 h. The final siRNA concentration was 100 nM in each well, and in order to avoid the toxic interference brought by ITZ, the content of ITZ in each formulation was controlled below 2 nM. After 4 h incubation, the transfection medium in each well was replaced by fresh medium containing 10% FBS and incubated for another 20 h. 

Finally, the total RNA was extracted from the cells with TRIzol Reagent (Invitrogen, Carlsbad, CA, USA), then transcribed to cDNA using ReverAid First Strand cDNA Synthesis kit (Fermentas, Maharashtra, India). The mRNA levels of the target genes were quantified by real time PCR using SYBR Green qPCR kit (Takara Biotechnology Co., Ltd., Dalian, China) along with the selected DNA primer pairs. Primer pairs used were VEGF (forward, 5′-GAAGACACGGTGGTGGAAGAAGAG-3′; reverse, 5′-GGGAAGGGAAGATGAGGAAGGGTA-3′) and GAPDH (forward, 5′-GAGCCAAAAGGGTCATCATCT-3′; reverse, 5′-AGGGGCCATCCACAGTCTTC-3′). All the results were expressed as x ± s of three measurements.

### 2.11. In Vitro Angiogenesis and Tube Formation

In vitro angiogenesis and tube formation assays were conducted to evaluate the ability of tubular formation of endothelial cells after treatment [14,31]. Briefly, 50 µL of liquefied Matrigel was placed in 96 well plates and incubated in 37 °C for 30 min. HUVEC cells in DMEM medium alone or with the various formulations of NPs (N/P = 30, C/N = 1/5; ITZ: 8.28%) at 100 nM VEGF siRNA were seeded onto the surface of the Matrigel at a final density of 1 × 10^4^ cells per well for 18 h in a 37 °C incubator containing 5% CO_2_.

To investigate 4T1-related endothelial tube formation and angiogenesis efficiency, the 4T1 cells were seeded into 6-well plates and cultured for 24 h at a density of 1 × 10^4^ cells per well [32]. Then, the cells were treated and transfected with different formulations of NPs (N/P = 30, C/N = 1/5; ITZ: 8.28%) at 100 nM VEGF siRNA and cultured for 24 h. Their media were called “conditioned medium”. Subsequently, HUVEC cells in “conditioned medium” were seeded onto the surface of the Matrigel at a final density of 1 × 10^4^ cells per well for 18 h in a 37 °C incubator containing 5% CO_2_. Final pictures were captured via IX51 inverted fluorescence microscope (Olympus Corporation, Tokyo, Japan).

### 2.12. Inhibition of the Effect of ITZ-siRNA^VEGF^ NPs in Cell Proliferation

4T1 cells were seeded in 96-well plates at the density of 3 × 10^3^ cells per well and incubated for 24 h to allow cell attachment. Naked siRNA (siRNA^VEGF^), ITZ, blank NPs (PEI-PLA/PEG-PAsp), siRNA^VEGF^ NPs, ITZ NPs and ITZ-siRNA^VEGF^ NPs (N/P = 30; C/N = 1/5; ITZ content: 8.28%) were added to the cells and incubated for 72 h. The final siRNA concentration was 20 nM in each well, and the optical density (OD) was measured at 450 nm using the Synergy H1m Monochromator-Based Multi-Mode Microplate Reader (BioTek., Dallas, TX, USA). Untreated cells were taken as control with 100% of viability. The results were expressed as x ± s of four measurements.

To study the cytotoxicity of ITZ, ITZ NPs and ITZ-siRNA^VEGF^ NPs, A549 cells and 4T1 cells were seeded at 5 × 10^3^ cells/well plate. After overnight incubation, the medium was replaced with fresh medium containing different concentrations of ITZ (0.05~5 µg/mL) or siRNA (20 nm per well) for 48 h. After the incubation, 10 μL of CCK-8 reagent was added to each well, cultured for 3 h, and the absorbance values of each well were measured at 450 nm. Untreated cells served as controls with 100% viability.

### 2.13. Cell Apoptosis Study

4T1 cells were seeded into a 12-well plate at a density of 5 × 10^4^ cells per well and treated with PBS, siRNA^VEGF^, ITZ, blank NPs, siRNA^VEGF^ NPs, ITZ NPs and ITZ-siRNA^VEGF^ NPs (N/P = 30; C/N = 1/5; ITZ content: 8.28%; 50 nM of siRNA/each well) for 24 h. For the quantitative measurement of apoptosis, the cells were harvested by 0.25% trypsin without EDTA, washed with PBS, resuspended in binding buffer, and stained with Annexin V-FITC/PI for 15 min and then analyzed with a FACSCalibur flow cytometer.

### 2.14. In Vivo Inhibitory Effect of ITZ-siRNA^VEGF^ NPs on Breast Cancer

Female BALB/c mice (4–6 weeks old, 18–22 g) were acquired from Vital River Laboratory Animal Technology Co., Ltd. (Beijing, China). The animal experiment was ethically approved by Laboratory Animal Ethics Committee in the Institute of Materia Medica in Peking Union Medical College. All the experimental procedures were performed in conformity with institutional guidelines and protocols for the care and use of laboratory animals.

A total of 1 × 10^5^ 4T1^Luc^ cells were orthotopically inoculated in the fourth mammary fat pad in the right lower abdomen of 4–6-week-old female BALB/c mice. When the tumor volume reached around 130 mm^3^, tumor-bearing mice were randomly assigned into five groups (*n* = 4/group). The mice were injected with the following preparations, respectively: saline, blank NPs, siRNA^VEGF^ NPs, ITZ NPs and ITZ-siRNA^VEGF^ NPs. ITZ was administered at a dose of 10 mg/kg, and VEGF siRNA was administered at a dose of 3 mg/kg. All the formulations were given to mice via tail vein every 3 days four times, and the tumor volumes were measured each 2 days (*n* = 4). After the final administration, 0.1 mL Luciferin (10 mg/mL) was intraperitoneally injected, and the mice were anesthetized with 1–2% isoflurane for 10–15 min. Then, the mice were fixed in the Xenogen in vivo imaging system to detect tumor bioluminescence. During the experiment, body weights and tumor sizes of the mice were determined.

### 2.15. Detection of VEGF and CD31 Expression in Tumor Tissues

Two weeks after the last administration, the mice were euthanized by cervical dislocation, after which, the tumor tissues were resected and fixed in 4% neutral formaldehyde solution for 72 h and subjected to VEGF mRNA expression. For the immunohistochemical analysis, the tumor sections were fixed with 4% neutral formaldehyde and were incubated with a monoclonal rabbit polyclonal anti-VEGF antibody (1:250) and anti-CD31 antibody (Abcam, Cambridge, UK, ab28364) at 4 °C overnight, respectively. The secondary antibody (goat antirabbit IgG-HRP) (Cell Signaling Technology, Inc (CST), Boston, MA, USA) was applied (1:1000) and incubated for 45 min at room temperature. The sections were visualized and photographed under a light microscope.

### 2.16. In Vivo Safety Evaluation

Two weeks after the last administration, blood samples were collected from the orbit, and the alanine aminotransferase (ALT) and aspartate aminotransferase (AST) levels were measured to evaluate drug toxicity in mice.

### 2.17. Statistical Analysis

Data are presented as the mean ± standard deviation (SD). Significant differences between two groups were evaluated using Student’s *t*-test. Comparisons among multiple groups were performed by one-way analysis of variance (ANOVA) with Bonferroni’s post hoc test.

## 3. Results

### 3.1. Synthesis and Characterization of PEI-PLA Copolymer

The synthesis of PEI-PLA was prepared by amino reaction between the carboxyl group of PLA-COOH and amino groups of PEI (Figure 2A). In the ^1^H-NMR spectrum (Figure 2B), the peak of PEI appears at around 2.6 ppm [33]. In PEI-PLA, a new broad peak appears in 2.3–3.4 ppm, which is attributed to the protons of methylene (–CH_2_CH_2_–) in PEI. The signals at δ = 1.20 ppm and δ = 4.08 ppm corresponded to the –CH_3_ and (–CH) proton in the PLA block of PEI-PLA, respectively. As shown in Figure 2C, strong absorption appears at 1752.6 cm^−1^ in the FTIR of PLA-COOH, which is attributed to the stretching vibration absorption peak (V_C=O_) of the carboxyl group in PLA-COOH. In PEI-PLA, strong absorption appeared at 3303.9 and 1540–1640 cm^−1^, of which 3303.9 cm^−1^ was attributed to the absorption peak of amino on PEI, and 1540–1640 cm^−1^ was attributed to the characteristic absorption peak of C=O stretching vibration in the amide bond. The results showed that the target molecule was successfully synthesized according to the reaction.

### 3.2. Characterization of the Nanoparticles

The nanocarriers with a core-shell copolymeric structure and PEGylation on the surface were formulated by a three-step method of dialysis for the core and complex coacervation method for siRNA loading and PEGylation (Figure 3A). PLA, which is a good hydrophobic and biodegradable copolymer, was chosen to encapsulate the extremely hydrophobic ITZ in the core of the ITZ-siRNA^VEGF^ NPs. In the previous study [4], the optimal carrier material ratio of N/P 30 and C/N ratio 1/5 was selected by orthogonal experiments. 

Thus, at this optimal ratio, we successfully formulated a nanocarrier. Particle size and zeta potential of Blank NPs measured by dynamic light scattering (DLS) were 105.4 ± 8.5 nm and 16.5 ± 3.21 mV (Figure 3B), while those of ITZ-siRNA^VEGF^ NPs were 117.9 ± 10.3 nm and 6.69 ± 2.46 mV (Figure 3C), respectively. The satisfactory PDI of ITZ-siRNA^VEGF^ NPs (PDI of 0.134 ± 0.072) indicates a narrow, uniform, homogenous distribution and successful development of the formulation. The morphology images of ITZ-siRNA^VEGF^ NPs from TEM (Figure 3B,C) further demonstrated that the NPs were spherical with smooth surface. 

The EE of ITZ was 90.26 ± 2.60%, and the DL was 8.28 ± 0.52%. The results of gel imaging also showed that, when the N/P ratio was 30 and C/N ratio was 1/5, VEGF siRNA could completely shrink. Even after ITZ was encapsulated in the hydrophobic core, the siRNA shrinkage was not affected (Figure 3D). In addition, we further accessed the stability of ITZ-siRNA^VEGF^ NPs in PBS buffer and PBS buffer containing 10% FBS, which is an indication of their aggregation behavior in vivo after systemic administration. This was evaluated by measuring their particle size changes over time. As shown in Figure 3E, under the conditions of two buffers, the particle size of ITZ-siRNA^VEGF^ NPs remained slightly changed within 6 h, and the change range of the particle size within 24 h was no more than 10 nm.

To confirm the pH-responsive sheddable ability of PEG-PAsp, we evaluated the particle size, zeta potential and PDI of ITZ-siRNA^VEGF^ NPs under different pH values. The results indicate that the size decreased from 123.1 ± 11.4 nm to 88.3 ± 9.2 nm, and the zeta potential increased from 6.6 ± 1.2 mV to 21.7 ± 5.6 mV as the pH value decreased from 8.0 to 5.0 (Table 1), which indicate that the PEG-PAsp copolymer detached from the NPs when the pH value was below 6.0. These results verify our hypothesis that the PEG-PAsp block will become neutral in endosome (pH 5.0–6.0) and therefore detach from the complex nanoparticles.

### 3.3. In Vitro Release of ITZ

A comparative drug release study of ITZ-siRNA^VEGF^ NPs was performed in PBS at pH 5.5 and pH 7.4, simulating the acidic endosome and the normal physiological environment, respectively (Figure 3F). At pH 5.5, after 72 h, the release of ITZ from ITZ-siRNA^VEGF^ NPs was about 80.74 ± 4.58; however, at pH 7.4, the accumulative release of ITZ from ITZ-siRNA^VEGF^ NPs was 60.43 ± 2.19. The NPs released ITZ more quickly in the acid environment likely due to the detachment of PEG-PAsp outer and faster degradation of the PLA core.

### 3.4. Cellular Uptake of Coumarin 6 (C6) and Cy3 siRNA Co-Loaded NPs in 4T1 Cells

As shown in Figure 4A, after transfecting for 4 h, when compared with free Cy3 siRNA or C6, relatively high co-localization spots of green C6 and red Cy3 siRNA were found in Cy3 siRNA NPs and C6 NPs groups, suggesting that the efficiency of drug delivery to 4T1 cells was greatly improved after modification by delivery vectors. In the Cy3 siRNA-C6 NPs group, red and green fluorescent generated yellow stains in the cytoplasm (Figure 4(Ae)), which suggests that both drugs are efficiently delivered into cells with the help of delivery vectors and that the nanoparticles enter cells most likely by the endocytosis mechanism.

Two-color flow cytometry was used to quantify the cellular uptake of various NPs, and similar results were further visualized. Compared to free Cy3 siRNA, both the Cy3 siRNA NPs and Cy3 siRNA-C6 NPs groups had the highest mean fluorescence intensity and had significant improvement (*** *p* < 0.001) (Figure 4B). Similarly, from Figure 4C, we can see that the C6 NPs and Cy3 siRNA-C6 NPs groups had the highest mean fluorescence intensity and had a significant difference when compared with the free C6 group (*** *p* < 0.001). In both C6 and Cy3 siRNA channels, the uptake rates in single drug-loaded NPs and double drugs-loaded NPs did not have significant differences.

### 3.5. Endosomal Escape

To further investigate whether the co-loaded NPs could escape from endosomes/lysosomes following cell internalization, the intracellular fluorescence distribution was also investigated with CLSM. After transfecting for 1 h, co-localization spots of green C6 and red endosomes/lysosomes were found as shown in Figure 5A (As the arrow shows, there was a yellowish overlap), and the signals gradually increased at 2 h(As the arrow shows, there was more yellow overlap), suggesting that the released siRNA^VEGF^-C6 NPs were entrapped within endosomes/lysosomes. At 4 h, the green C6 gradually began to escape from the red endosomes/lysosomes (As the arrow shows, the yellow overlap gradually diminished, exposing the green of C6), until at 6 h, most of the C6 had successfully escaped (See the arrow). In Figure 5B, the red siRNA^Cy5^ gradually entered the green endosomes/lysosomes, and the yellow signal was the strongest at 4 h. 

More siRNA^Cy5^ dots were gradually separated from green fluorescence after 6 h incubation, thus, indicating that siRNA^Cy5^-ITZ NPs could achieve lysosomal escape into the cytoplasm at 6 h (As the arrow shows, most of the red siRNA had successfully escaped from the lysosome). These results demonstrated similar endosomal escape ability of two co-loaded drugs, which could support the evidence for that the composite nanoparticles are likely to enter the cell through endocytosis and subsequently escape from the endosomes/lysosomes through the proton sponge effect of PEI under acidic conditions.

### 3.6. Wound Healing Assay

Since cancer cell metastasis is associated with cell migration, we focused on the invasiveness of 4T1 breast cancer cells. The inhibition of tumor cell migration by ITZ-siRNA^VEGF^ NPs was investigated by wound healing assay. As shown in Figure 6A, 48 h after transfection, scratches without any treatment almost completely healed, indicating that 4T1 cells had the ability to repair scratches through plane migration. In the free siRNA and blank NPs groups, the scratch site basically healed after 48 h. The scratches in the siRNA^VEGF^ NPs, ITZ, ITZ NPs and ITZ-siRNA^VEGF^ NPs groups did not heal after 48 h, and the scratches in the ITZ-siRNA^VEGF^ NPs group even widened.

### 3.7. VEGF Silencing Efficiency and Anti-Angiogenesis Effect of ITZ-siRNA^VEGF^ NPs

As shown in Figure 6B, when compared with the untreated control group, there was no significant difference between the free siRNA and Blank NP groups. However, the mRNA expression of siRNA^VEGF^ NPs and ITZ-siRNA^VEGF^ NPs both exhibited good silencing effects, whereas free VEGF siRNA showed poor suppression because it was difficult to penetrate into the cells and easily degraded in the biological media. Here, siRNA was successfully transfected into cells with the help of PEI-PLA/PEG-PAsp delivery system to achieve silencing effect. Compared with the control group, the expression of VEGF mRNA in ITZ NPs was also decreased, which may be related to the inhibitory effect of ITZ on vascular growth. Studies have shown that ITZ can inhibit the formation of micro vessels, and the possible mechanism is that ITZ inhibits vascular cell growth factors such as VEGF, AAMP and e-nos [34]. Consistent with the above theories, among the results of this experiment, ITZ-siRNA^VEGF^ NPs exhibited the best gene silencing efficiency.

The formation and development of endothelial cell capillary structure is a multistep process, which involving cell adhesion, migration, differentiation and growth [31]. Cancer cells can secrete various inducers into the microenvironment to regulate angiogenesis [32], we investigated the in vitro antiangiogenesis of HUVEC cells when nanoparticles were directly treated with HUVEC cells and when nanoparticles were cultured with 4T1 cells. First, to assess the direct angiogenetic inhibitory effect of the various nanoparticles, HUVECs were co-cultured with the nanoparticles and plated onto Matrigel for 24 h. 

Compared with the control group (Number of Nodes: 146 ± 8.54; Total length: 4881 ± 136.27), the anti-tube formation and antiangiogenesis ability of siRNA^VEGF^, siRNA^VEGF^ NPs, ITZ, ITZ NPs and ITZ-siRNA^VEGF^ NPs groups was clearly increased, among which, our final preparation group of siRNA^VEGF^ NPs (Number of Nodes: 0 ± 0; Total length: 111 ± 14) had the highest efficacy, with a significant difference (** *p* < 0.01). The effect of 4T1 cells’ VEGF downregulation on the endothelial tube formation of HUVECs was also investigated. 

The conditioned medium treated with different nanoparticles was collected and cultured with HUVECs for 24 h. Similar to the previous experiment, our final preparation group of siRNA^VEGF^-ITZ NPs had the highest anti-tube formation and antiangiogenesis ability (Number of Nodes: 7.67 ± 2.08; Total length: 1124.67 ± 152.16) on HUVECs after pre-treated with 4T1 cells (Figure 6(Cb),D,E). Consistent with the results of VEGF gene silencing experiment, vector-modified siRNA showed better inhibition efficiency than free siRNA, whether directly acting on HUVECs or after incubation with 4T1 cells.

### 3.8. Cell Inhibition and Apoptosis Analysis

In order to investigate the effects of ITZ and VEGF siRNA on cell proliferation, CCK-8 method was used to measure the cell inhibition rate of different nanoparticles. As shown in Figure 7A, compared to the control group, cell proliferation inhibitory rates of ITZ, ITZ NPs, and ITZ-siRNA^VEGF^ NPs treated groups were significantly elevated (** *p* < 0.01) in both 4T1 and A549 cell lines. Within a certain concentration range, ITZ-siRNA^VEGF^ NPS with dual loading showed the highest inhibition rate. For the 4T1 cell line, the survival rates of the cells treated with siRNA^VEGF^ NPs and ITZ NPs were 82.27% ± 5.65% and 55.85% ± 4.83%, respectively, while that of cells treated with ITZ-siRNA^VEGF^ NPs was 51.72% ± 4.62%. 

For the A549 cell line, the survival rates of the cells treated with siRNA^VEGF^ NPs and ITZ NPs were 80.59% ± 4.32% and 63.25% ± 6.15%, respectively, while that of cells treated with ITZ-siRNA^VEGF^ NPs was 59.58% ± 7.26%. ITZ was reported to induce cancer cell death via apoptosis mainly due to alteration of mitochondria membrane potential, reduction of Bcl-2 expression and increase of caspase-3 activity [9,35]. The greater cytotoxic effect of ITA-loaded NPs as compared to free ITZ against 4T1 cells could be attributed to the small size and rapid uptake of nano-formulations, which facilitates different mechanisms of transport, such as endocytosis or passive transport.

We further measured the cell inhibition rate of ITZ NPs and ITZ siRNA^VEGF^ NPs with different ITZ concentrations (0.05–5 µg/mL) by CCK-8 method. In Figure 7B, all formulations displayed a typical dose-dependent cytotoxicity to the 4T1 and A549 cells for 48 h. Free ITZ displayed certain antitumor activity, and this cytotoxicity was enhanced by delivering ITZ and VEGF siRNA using PEI-PLA/PEG-PAsp NPs to enhance cellular accumulation.

To study the apoptosis effect of ITZ-siRNA^VEGF^ NPs, we evaluated cell apoptosis post-drug treatment by flow cytometry, using an annexin V/FITC kit. As shown in Figure 7C, 4T1 cells treated with blank NPs (PEI-PLA/PEG-PAsp) showed 4.66% of apoptosis percentage, confirming that the blank polymeric delivery system produced minor effect on normal cell progression. The delivery of VEGF siRNA into 4T1 cells with PEI-PLA/PEG-PAsp NPs led to 12.22% cell apoptosis, while only 4.75% cell apoptosis was observed when the cells were treated with free siRNA. Similarly, the apoptosis rate of cells treated with ITZ NPs was increased to 13.72%, while the apoptosis rate of cells treated with free ITZ was 10.98%. However, the dual-loaded ITZ-siRNA^VEGF^ NPs led to the highest cell apoptosis rate of 14.16%.

### 3.9. In Vivo Antitumor Efficacy and Angiogenesis Suppression

We established an animal model of 4T1 breast cancer in situ and investigated the inhibition of ITZ-siRNA^VEGF^ NPs on the growth of tumor and the inhibition of VEGF expression. In Figure 8A,B, the luminescence intensity of the control group was above 6.5 × 10^6^ p/s/cm^2^/Sr, and the luminescence intensity was the strongest, followed by the blank nano group, which was above 5.8 × 10^6^ p/s/cm^2^/ Sr. 

The intensities of ITZ NPs and ITZ-siRNA^VEGF^ NPs were about 2.0 × 10^6^ p/s/cm^2^/Sr and 1.7 × 10^6^ p/s/cm^2^/Sr, respectively, with significant differences when compared with the control group (** *p* < 0.01). In Figure 8C,D, the tumor size of control group and blank NPs group continued to increase. These increased from 98.27 ± 22.57 mm^3^ and 105.92 ± 15.80 mm^3^ to 345.71 ± 76.88 mm^3^ and 331.68 ± 83.27 mm^3^, respectively. siRNA^VEGF^ NPs and ITZ-NPs groups also increased from 97.62 ± 21.947 mm^3^ and 106.32 ± 15.76 mm^3^ to 251.86 ± 37.15 mm^3^ and 193.67 ± 48.49 mm^3^, respectively. ITZ-siRNA^VEGF^ NPs showed the best tumor inhibition, growing from 106.42 ± 15.72 mm^3^ to 142.91 ± 23.62 mm^3^, respectively. 

The histological examination of the tumor sections for apoptosis and necrosis was performed after H&E and TUNEL staining, as shown in Figure 9A. H&E staining results showed that the normal saline and blank NPs groups had the typical histological features of tumor cells with less necrosis. In the siRNA^VEGF^ NPs and ITZ NPs groups, extensive focal necrosis of cancer cells and pathological mitosis could be observed, and the ITZ-siRNA^VEGF^ NPs group showed more extensive necrosis and fragmented nucleus with less viable tumor cells regions. In the TUNEL assay (Figure 9A,B), 53.26 ± 7.26% of apoptotic cells was observed in the co-loaded ITZ-siRNA^VEGF^ NPs group, which was much higher than that in the single-loaded groups (siRNA^VEGF^ NPs: 12.45 ± 5.11%; ITZ NPs: 22.45 ± 4.85%).

Subsequent to the in vitro investigations, the anti-angiogenic effect of anti-VEGF and anti-CD31 developed ITZ-siRNA^VEGF^ NPs was investigated in the 4T1 breast cancer-bearing mice to study the in vivo angiogenesis suppression. As shown in Figure 9B, compared with the control group (2.10 ± 0.19) and Blank NPs group (1.91 ± 0.17), VEGF mRNA relative expression of siRNA^VEGF^ NPs (0.90 ± 0.18), ITZ NPs (0.83 ± 0.17) and ITZ-siRNA^VEGF^ NPs (0.67 ± 0.07) decreased significantly (** *p* < 0.01). The immunohistochemical results showed that the expression levels of VEGF and CD31 in the ITZ-siRNA^VEGF^-NPs treatment groups were significantly suppressed than those of the control group (Figure 9A,B; ** *p* < 0.01). CD31 is an endothelial marker for quantifying angiogenesis. The results were consistent with the obtained mRNA expression levels of VEGF and in vitro anti-angiogenesis study.

### 3.10. In Vivo Safety

In order to investigate the in vivo toxicity of NPs and vectors, we recorded the changes of body weight and liver function indexes during the experiment. Figure 10A showed that the mice in each experimental group did not show significant weight loss during the drug administration cycle, and there was no significant difference among each group. 

As it has been reported that ITZ could induce hepatic dysfunction, after four consecutive intravenous injections at 10 mg/kg dose every 3 days and an observation period, we determined serum ALT and AST levels of the mice. As shown in Figure 10B,C, there was no significant differences of ALT and AST levels in the Blank NPs, siRNA^VEGF^ NPs, ITZ NPs or ITZ-siRNA^VEGF^ NPs. More than that, the ALT and AST levels of ITZ-siRNA^VEGF^ NPs group even decreased when compared with the control group (* *p* < 0.05).

## 4. Discussion

Many researchers believe that weakly positive nanoparticles in the appropriate particle size range can be less readily absorbed, and make the best use of EPR effect [36,37]. So the slightly positive zeta potential after PEGylation of the ITZ-siRNA^VEGF^ NPs reduces the chances of interaction of nanoparticles with phagocytes and their absorption by these cells. This shielding effect of PEG enhances the stability of the NPs especially, and also helps to improve circulation time in vivo by shielding recognition by the reticular endothelial system (RES) in the body [14,38]. 

ITZ is an extremely poorly water-soluble (~1 ng/mL in water) molecule with high lipophilicity (log P 5.66) [39,40]. Herein, we encapsulated ITZ into nano-sized suspension to increase the solubility to enable intravenous administration. The ITZ-siRNA^VEGF^ NPs presented good pH-sensitive property and sustained drug release profile. Such behavior presumably enhances intracellular drug release once the complex NPs enter the tumor cells via endocytosis and trapped within the acidic endosomal compartments [41,42]. The drug release from polymeric nanocarrier systems involves several mechanisms such as polymer degradation, erosion of the polymer, desorption from the particle surfaces [43,44]. The initial rapid release and subsequent controlled release would maintain the effective concentration of ITZ in PBS for a long time.

As a ligand produced by tumor cells and associated stroma, VEGF can activate multiple downstream pathways, results in endothelial cell proliferation and migration [45,46]. Rudin and colleagues [47] investigated that ITZ could inhibit cell migration, chemotaxis and tube formation on human umbilical vein endothelial cells. In this study, under the combined action of VEGF siRNA and ITZ, the ITZ siRNA^VEGF^ NPs group significantly inhibited the plane migration ability of cells and inhibited cell wound healing, which further confirmed the antimigration ability of VEGF. 

VEGF is considerably expressed in the metastatic stages of cancer, especially breast cancer; therefore, the VEGF promoter can be an appropriate promoter for the transcriptional targeting [48]. ITZ directly [8] or downregulate VEGF via VEGF siRNA [31] is a promising strategy for cancer therapy, which can result in the inhibition of tumor angiogenesis and metastasis. First, we investigated whether ITZ-siRNA^VEGF^ NPs could efficiently knockdown the expression of the therapeutic target gene VEGF with 4T1 breast cancer cell. Previous reports have suggested that ITZ can induce cell death via apoptosis induction in breast cancer cells [49]. Furthermore, it had been reported that VEGF provides a survival signal for breast tumor cells in vitro and blockade of VEGF results in apoptosis of these cells [50]. From the result, it can be speculated that the co-delivery of ITZ and VEGF siRNA activated the intrinsic apoptotic pathway, promoted cell apoptosis and played a synergistic efficacy.

We established an animal model of 4T1 breast cancer in situ and investigated the inhibition of ITZ-siRNA^VEGF^ NPs on the growth of tumor and the inhibition of VEGF expression. Accordingly, the weight of tumor tissue in each group showed that the tumor weight in the ITZ-siRNA^VEGF^ NPs group was the lightest. The growth of tumor volume in mice treated with ITZ-siRNA^VEGF^ NPs was well inhibited, indicating that the antitumor effect of ITZ and VEGF siRNA on 4T1 breast cancer mice was clear. The HE and TUNEL stained tumor sections revealed that the co-loaded NPs treatment induced significant cell apoptosis and necrosis of the tumor tissues when compared with the control group, indicating the excellent therapeutic efficacy of ITZ-siRNA^VEGF^ NPs. 

ITZ itself can inhibit mTOR and VEGF2 simultaneously by inhibiting the operation of cholesterol, while VEGF siRNA can directly silence the highly expressed VEGF gene in tumor cells. Through the synergism of these two drugs, the growth of tumor blood vessels can be inhibited, further suppressing the growth and development of tumor. To conclude, ITZ-siRNA^VEGF^-NPs could reduce VEGF and CD31 expression through the synergistic effect of ITZ and VEGF siRNA and inhibited growth and angiogenesis of breast cancer in vivo. 

Mice that were implanted with tumor cells caused liver damage, which was recovered after treatment with ITZ-siRNA^VEGF^ NPs, indicating that our nano-delivery systems did not induce liver cytotoxicity in mice. Although ITZ is not the most effective antitumor drug compared to other powerful antitumor drugs, due to its low toxicity and good antiangiogenic anticancer activity [8], ITZ has been chosen for several clinical trials with cancer patients [51].

## 5. Conclusions

In conclusion, we successfully developed a multifunctional pH-sensitive platform for combined tumor chemotherapy and gene therapy using ITZ and VEGF siRNA as model drugs. The prepared nanoparticles showed a proper particle size, narrow distribution, weakly positive surface charge, high drug loading, good in vitro stability, and controlled drug release. The carrier-modified nanoparticles showed higher cellular uptake efficacy and gene silencing efficiency than the free drugs. When compared with free drugs or single-loaded nanoparticles, the co-loaded nanoparticles showed the highest cytotoxicity, apoptotic cell death, anti-angiogenesis effect and anti-migration efficiency in the cancer cell lines. 

The in vivo results demonstrated that the co-loaded ITZ-siRNA^VEGF^ NPs could inhibit tumor growth effectively due to the combined anti-angiogenesis and anti-tumor effect, as well as low toxicity and little side effects. Overall, our approach is a promising and applicable treatment strategy of nanocarriers for effective targeted anticancer drugs and siRNA drugs. Accordingly, PEI-PLA/PEG-PAsp vectors offer promise as a bioactive nano-platform for the co-delivery of other poorly water-soluble drugs and gene drugs. 

In terms of the total vessel length, the inhibition effect of siRNA^VEGF^ NPs was even higher than that of free ITZ or ITZ NPs, indicating that VEGF siRNA with the help of the vectors can further successfully inhibit the generation and development of blood vessels in vitro by down-regulating the expression of VEGF-related genes. This result could be attributed to the distinctive performance of the PEI-PLA/PEG-PAsp-based delivery system with the excellent enhancement of cellular uptake and gene transfection efficiency as mentioned above.

## Figures and Tables

**Figure 1 pharmaceutics-14-01369-f001:**
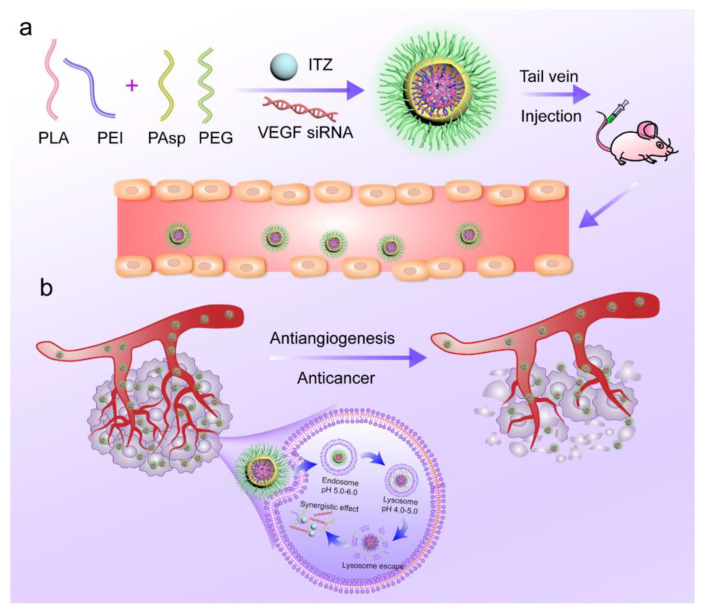
(**a**) ITZ-VEGF siRNA-loaded composite nanoparticle delivery system. (**b**) Schematic of the intracellular therapeutic mechanism of the ITZ-VEGF siRNA composite nanoparticles.

**Figure 2 pharmaceutics-14-01369-f002:**
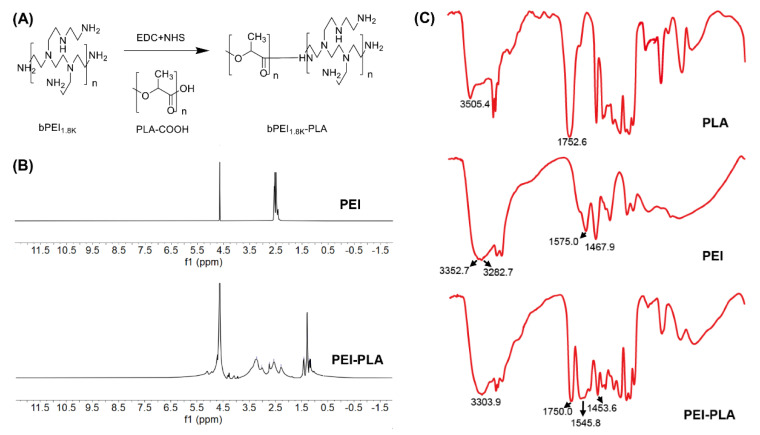
(**A**) Synthesis of PEI-PLA. (**B**) 1H-NMR spectra of PEI and PEI-PLA. (**C**) FTIR spectra of PLA-COOH and PEI-PLA.

**Figure 3 pharmaceutics-14-01369-f003:**
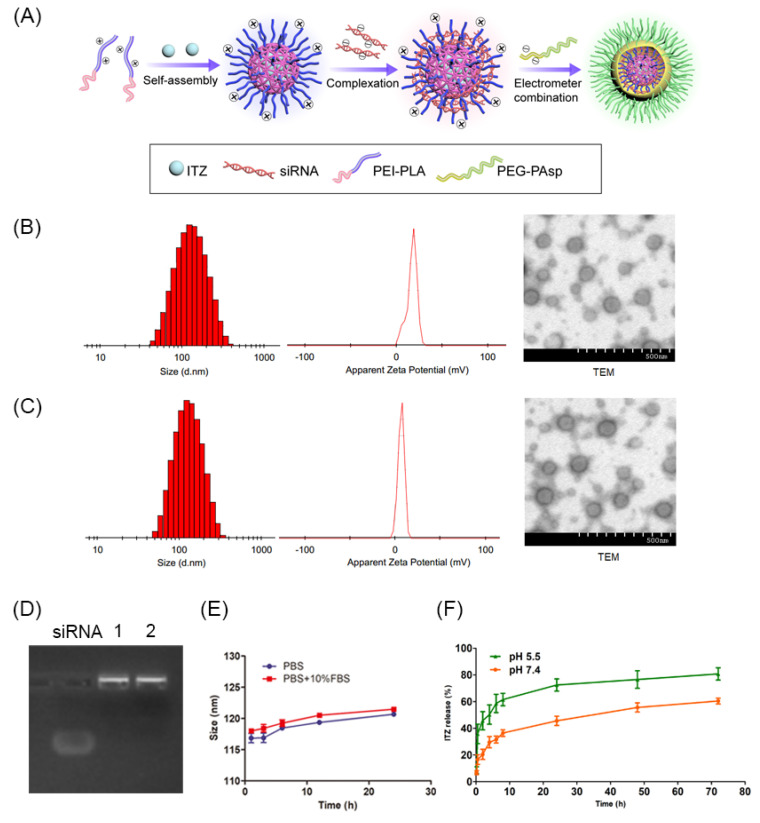
(**A**) ITZ/siRNA-loaded layer-by-layer nanoparticle delivery system. Particle size, zeta potential and TEM of Blank NPs (**B**) and ITZ-siRNA^VEGF^ NPs (**C**). (**D**) Gel electrophoresis assay. (**E**) Time-dependent colloidal stability of ITZ-siRNA^VEGF^ NPs. (**F**) Cumulative in vitro release profiles of ITZ-siRNA^VEGF^ NPs in PBS release medium at pH 5.5 and pH 7.4, respectively. Data plots and error bars represent the mean ± SD (*n* = 3), (N/P = 30, C/N = 1/5 and ITZ content of 8.28%).

**Figure 4 pharmaceutics-14-01369-f004:**
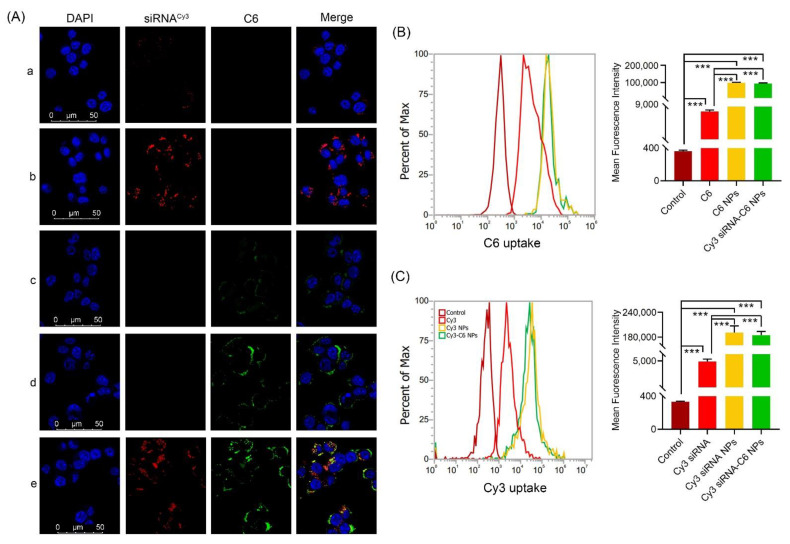
Cellular uptake of C6 or Cy3 siRNA in different formulations. 4T1 cells were analyzed after 4 h of incubation at the final concentrations of different NPs (C6 content 100 ng/mL, Cy3 siRNA 50 nM, N/P = 30 and C/N = 1/5). (**A**) CLSM images of cellular uptake of C6-siRNA^Cy3^ NPs. (a) siRNA^Cy3^; (b) siRNA^Cy3^ NPs; (c) ITZ; (d) ITZ NPs; and (e) siRNA ^Cy3^-ITZ NPs. For each column, from left to right: nuclei were stained by DAPI (blue); C6 fluorescence in cells (green); Cy3 signal in cells (red); merged with nucleus, C6 and siRNA^Cy3^. Scale bar = 50 µm. (**B**) Quantitative analyses of C6 uptake by flow cytometry. (**C**) Quantitative analyses of Cy3 siRNA uptake by flow cytometry. *** *p* < 0.001 as compared with C6 or Cy3 siRNA (*n* = 3).

**Figure 5 pharmaceutics-14-01369-f005:**
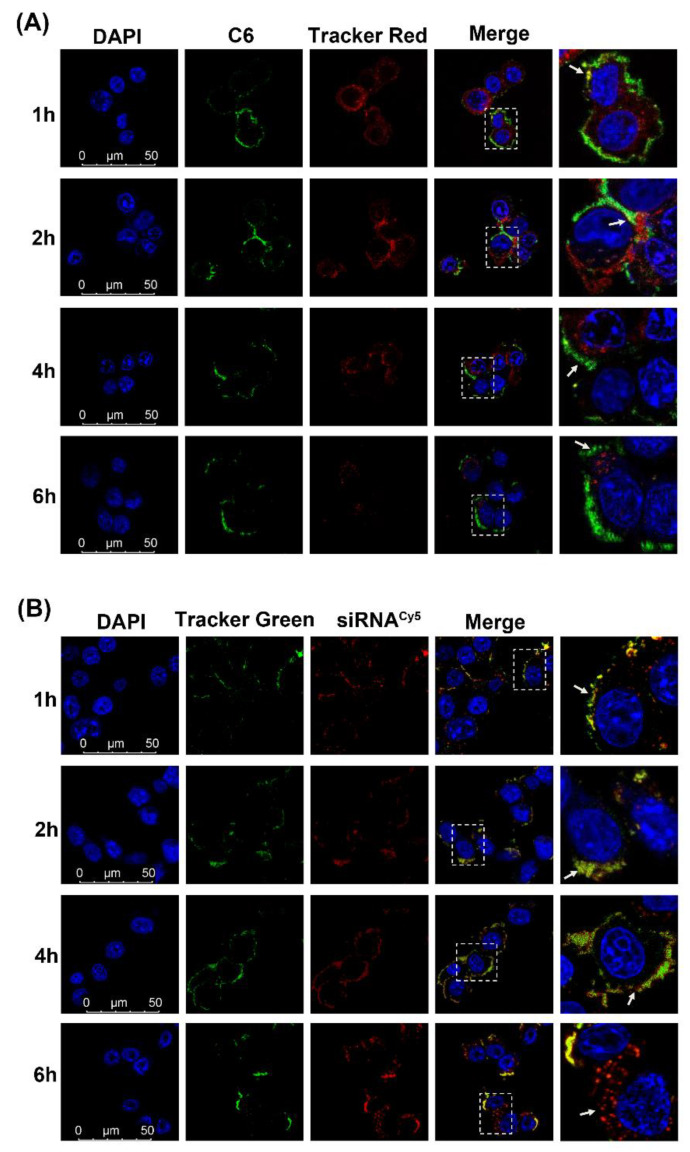
Endosomal escape of C6-siRNA^VEGF^-NPs (**A**) and Cy5 siRNA-ITZ NPs (**B**) in 4T1 cells undergoing 1, 2, 4 and 6 h, respectively. In order to reduce the ITZ influence, the concentration of ITZ is as low as 1 nM. DAPI for nuclei staining (blue), C6 (green), Cy5 siRNA (red), LysoTracker Red (Tracker Red) and LysoTracker Green (Tracker Green) for endosomes (red and green) were recorded. Scale bar = 50 µm.

**Figure 6 pharmaceutics-14-01369-f006:**
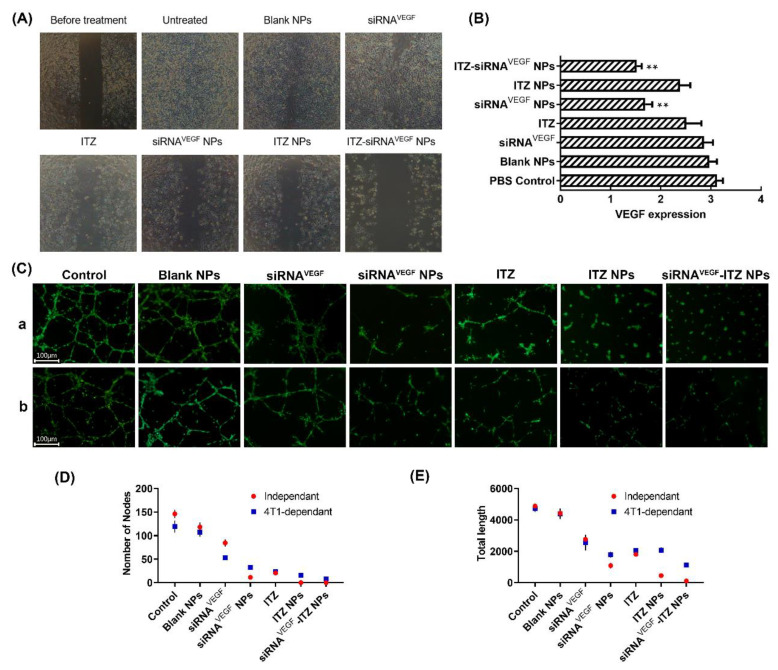
(**A**) Wound healing assay of different Nps (N/P = 30, C/N = 1/5, ITZ content: 8.28%, 100 nM of siRNA per well) on 4T1 cells. The healing situation of scratch wound was observed 48 h after scratching using an inverted microscope. Cells without any treatment were used as control. (**B**) Expression of VEGF mRNA determined by quantitative real-time PCR after 48 h of incubation at the final concentrations of different NPs (N/P = 30, C/N = 1/5, ITZ content: 8.28%, 100 nM of siRNA in per well). ** *p* < 0.01 as compared with the control (*n* = 3). (**C**) Representative fluorescent images of In vitro anti-angiogenesis efficacy of the nanoparticles. (a) Inhibition of tubule formation in HUVECs on matrigel after treatment of different NPs (N/P = 30, C/N = 1/5, ITZ content: 8.28%, 100 nM of siRNA per well) for 24 h. (b) Inhibition of tubule formation in HUVECs on matrigel after treatment of 4T1 cells-treated “conditioned medium” for 24 h. (**D**). The quantitative analysis of the inhibition of node numbers. (**E**) The quantitative analysis of the inhibition of total length. The results are represented as the mean ± SD, ** *p* < 0.01 as compared with the control (*n* = 3).

**Figure 7 pharmaceutics-14-01369-f007:**
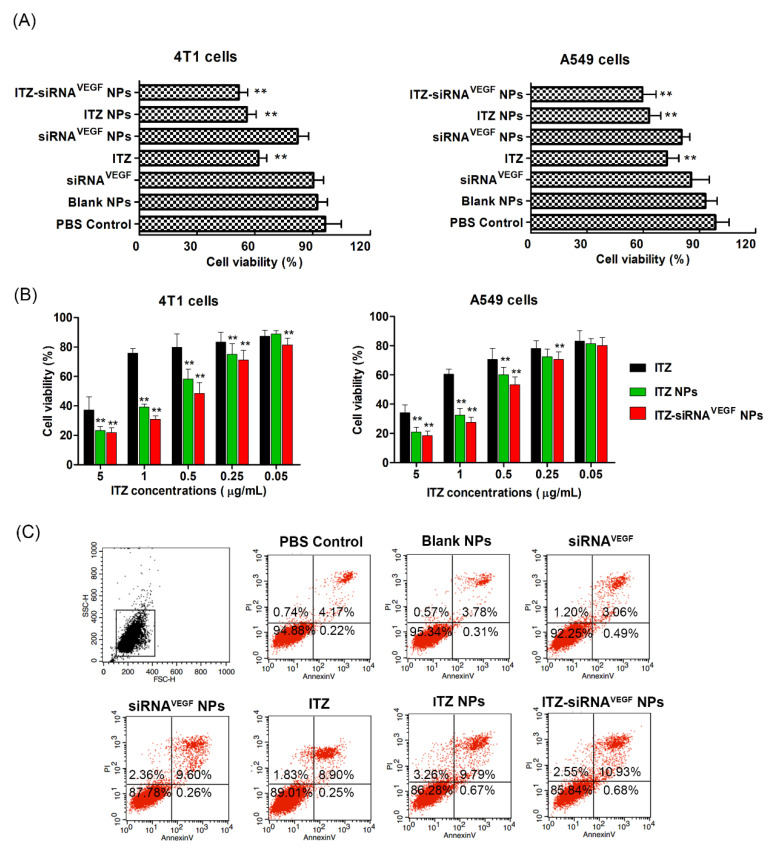
(**A**) In vitro cytotoxicity of 4T1 cells and A549 cells treated with different NPs for 48 h (N/P = 30, C/N = 1/5, ITZ content: 8.28%, 20 nM of siRNA in per well). ** *p* < 0.01 Vs PBS Control, Blank NPs and siRNA^N^.^C^ NPs (*n* = 3). (**B**) Cell viability of 4T1 cells and A549 cells after treated with different NPs. The concentration of ITZ varied from 0.05 to 5 µg/mL, and the concentration of both the scrambled siRNA and VEGF siRNA was 20 nM. ** *p* < 0.01 as compared with ITZ group (*n* = 3). (**C**) Cell apoptosis on 4T1 cells 24 h after treating with various NPs (N/P = 30, C/N = 1/5, ITZ content: 8.28%, 50 nM of siRNA per well). The cells were stained with Annexin V-FITC/PI for 15 min.

**Figure 8 pharmaceutics-14-01369-f008:**
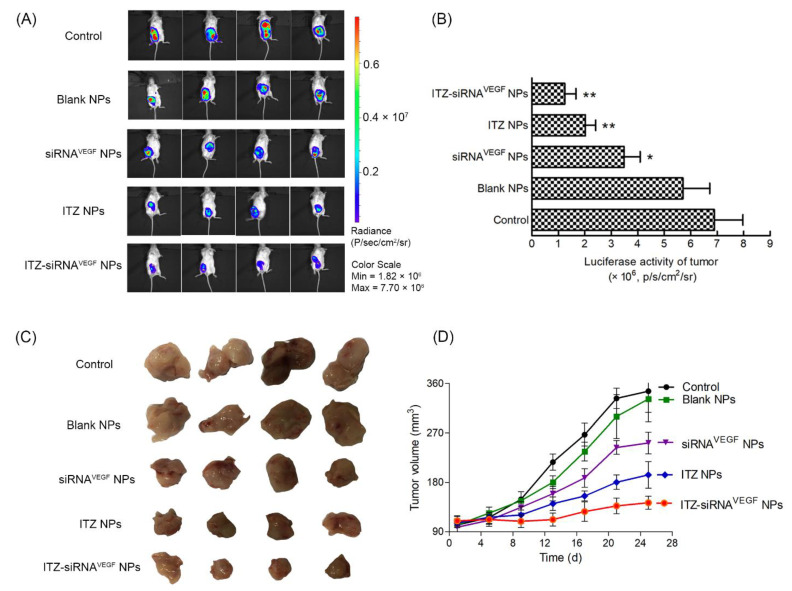
In vivo anti-tumor effect of systemic administration of ITZ and VEGF siRNA co-delivering NPs in 4T1 tumor-bearing mice. Female BALB/c mice (4–6 weeks old, 18–22 g) in each group were orthotopically inoculated with 1 × 10^5^ 4T1^Luc^ cells in the fourth mammary fat pad in the right lower abdomen. The mice were intravenously injected with various formulations (N/P = 30, C/N = 1/5; ITZ content: 8.28%). ITZ was administered at a dose of 10 mg/kg, and VEGF siRNA was administered at a dose of 3 mg/kg. (**A**) In vivo bioluminescence imaging analysis of mice after administration with various formulations. (**B**) Quantitative estimation by bioluminescent analysis. * *p* < 0.05, ** *p* < 0.01 vs. control group (*n* = 3). (**C**) The tumor photos (*n* = 4). (**D**) The changes in tumor volume due to in vivo antitumor effects of ITZ and VEGF siRNA co-delivering NPs (*n* = 4).

**Figure 9 pharmaceutics-14-01369-f009:**
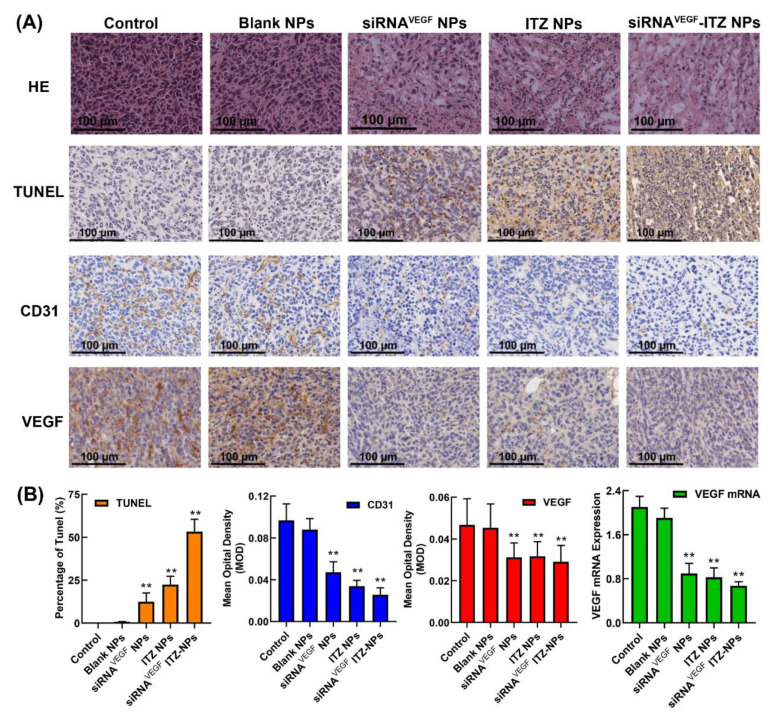
(**A**) Histological analysis of the tumor tissues after treatment. The HE, TUNEL staining, and immunofluorescence of CD31 and VEGF proteins in tumor tissues harvested from different groups. Scale bar = 100 µm. (**B**) Apoptosis rate (Percentage of TUNEL), quantitate analysis of CD31 and VEGF expression and tumor levels of VEGF mRNA after systemic administration of ITZ and VEGF siRNA co-delivery NPs (N/P = 30, C/N = 1/5; ITZ content: 8.28%) in 4T1^Luc^ tumor-bearing mice. ** *p* < 0.01 vs. saline (*n* = 4).

**Figure 10 pharmaceutics-14-01369-f010:**
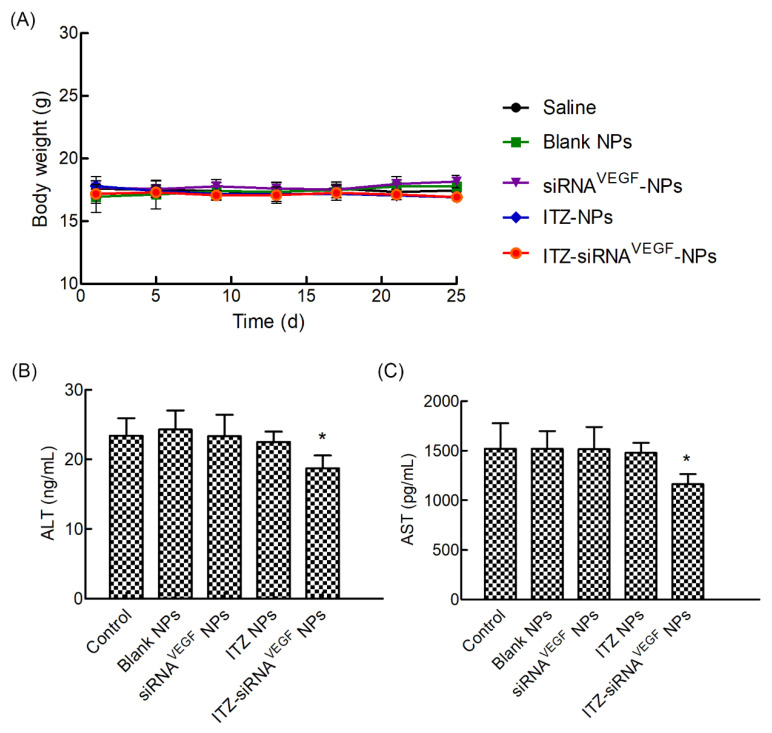
(**A**) The changes of body weight of in vivo anti-tumor effects of systemic administration of ITZ and VEGF siRNA codelivery NPs in 4T1^Luc^ tumor-bearing mice. (*n* = 4). (**B**) Mouse ALT levels in the serum of the tumor-bearing mice at the end time point of the animal experiment were determined by ELISA (*n* = 4). Data are provided as the mean ± SD. * *p* < 0.05, vs. saline. (**C**) Mouse AST levels in the serum of the tumor-bearing mice at the end time point of the animal experiment were determined by ELISA (*n* = 4). Data are provided as the mean ± SD. * *p* < 0.05, vs. saline.

**Table 1 pharmaceutics-14-01369-t001:** Particle sizes and zeta potentials of ITZ-siRNA^VEGF^ NPs at different pH values revealed pH-dependent attachment and detachment of PEG-PAsp under acidic conditions.

ITZ-siRNA^VEGF^ NPs	Size (nm)	Zeta Potantial (mV)	PDI
pH 5.0	88.3 ± 9.2	21.7 ± 5.6	0.203 ± 0.014
pH 6.0	90.2 ± 7.5	15.3 ± 3.0	0.211 ± 0.008
pH 7.0	115.4 ± 12.3	7.4 ± 2.1	0.156 ± 0.012
pH 8.0	123.1 ± 11.4	6.6 ± 1.2	0.152 ± 0.014

## Data Availability

All data available are reported in the article.

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
