# Peer review of "Co-Delivery of Repurposing Itraconazole and VEGF siRNA by Composite Nanoparticulate System for Collaborative Anti-Angiogenesis and Anti-Tumor Efficacy against Breast Cancer"

_pharmaceutics, 2022, doi:10.3390/pharmaceutics14071369_

Round 1

Reviewer 1 Report

I am no expert in the development this type of methodology but I found the paper very clear, well written and allowing a non-expert, like me to follow and understand the development of the described Nano Particles.

As far as the discussion of the effects of VEGF inhibition is concerned, it would be useful to mention that, in some "in vivo" models (e.g. glioblastoma models) inhibition of VEGF eventually can lead to actual increase in motility  and its inhibition is  not always beneficial.

As far as the Hallmarks of Cancer is concerned, the authors may want to quote the most recent  update (Hanahan D.  Cancer Discovery 12:31, 2022) in which :inducing angiogenesis" is changed to "Inducing or accessing vasculature" following the discovery of non-angiogenic tumours that co-opt pre existing vessels (cfr also  Donnem et al Nat Rev Cancer 18:323, 2018 and Zhang Y. et al Angiogenesis 23:17, 2020). 

Author Response

First of all, I would like to express my heartfelt thanks to you for your valuable opinions and suggestion, and we have studied the above literatures carefully and quoted the three literatures in the introduction section of our revised manuscript.

Reviewer 2 Report

The manuscript is well written and experiments are well thought out.

I have only one question. The authors use ITZ plus VEGF-siRNA to check for wound healing assay (and several other read-out assays). Would adding external VEGF overcome their formulation-mediated decreased wound healing. These rescue experiments would confirm that the observed effect is directly mediated by VEGF. 

Author Response

Thank you very much for your recognition and valuable comments on my article. In this wound healing assay, VEGF and ITZ play a synergistic role in inhibiting migration under different mechanisms. VEGF siRNA silences related gene expression on tumor cell surface, while ITZ inhibits cell growth and reproduction. Here, the reviewer came up with it that would adding external VEGF overcome their formulation-mediated decreased wound healing. The reviewer's point of view is also very modest and important. However, in this article, we focus on the effectiveness of the combined delivery of VEGF and ITZ to see if the combined delivery is superior to the single delivery. Therefore, we didn’t do further in-depth study on the intervention and mechanism of VEGF on scratch healing. If we carry out a deeper mechanism discussion on this topic in the future, we will further verify this part according to the reviewer's suggestion. We really appreciate to the reviewers for their valuable advice.